# Clinical Post-SARS-CoV-2 Infection Scenarios in Vaccinated and Non-Vaccinated Cancer Patients in Three German Cancer Centers: A Retrospective Analysis

**DOI:** 10.3390/cancers14153746

**Published:** 2022-07-31

**Authors:** Evgenii Shumilov, Lena Aperdannier, Nicole Schmidt, Christoph Szuszies, Albrecht Neesse, Petra Hoffknecht, Cyrus Khandanpour, Jan-Henrik Mikesch, Matthias Stelljes, Göran Ramin Boeckel, Phil-Robin Tepasse, Lea Reitnauer, Raphael Koch, Justin Hasenkamp, Ulrike Bacher, Simone Scheithauer, Lorenz Trümper, Norbert Schmitz, Gerald Wulf, Andrea Kerkhoff, Georg Lenz, Carolin Krekeler, Annalen Bleckmann

**Affiliations:** 1Department of Medicine A, Hematology, Oncology and Pneumology, University Hospital Münster, 48149 Münster, Germany; cyrus.khandanpour@ukmuenster.de (C.K.); jan-henrik.mikesch@ukmuenster.de (J.-H.M.); matthias.stelljes@ukmuenster.de (M.S.); leaelisabeth.reitnauer@ukmuenster.de (L.R.); norbert.schmitz@ukmuenster.de (N.S.); andrea.kerkhoff@ukmuenster.de (A.K.); georg.lenz@ukmuenster.de (G.L.); carolin.krekeler@ukmuenster.de (C.K.); 2Department of Hematology and Medical Oncology, University Medicine Göttingen (UMG), 37077 Göttingen, Germany; lena.aperdannier@med.uni-goettingen.de (L.A.); nicole.schmidt@med.uni-goettingen.de (N.S.); christoph.szuszies@med.uni-goettingen.de (C.S.); raphael.koch@med.uni-goettingen.de (R.K.); justin.hasenkamp@med.uni-goettingen.de (J.H.); lorenz.truemper@med.uni-goettingen.de (L.T.); gerald.wulf@med.uni-goettingen.de (G.W.); 3Department of Gastroenterology, Gastrointestinal Oncology and Endocrinology, University Medicine Göttingen (UMG), 37077 Göttingen, Germany; albrecht.neesse@med.uni-goettingen.de; 4Department of Thorax Oncology, Franziskus-Hospital Harderberg, Niels-Stensen-Kliniken, 49124 Georgsmarienhütte, Germany; petra.hoffknecht@niels-stensen-kliniken.de; 5Department of Hematology and Medical Oncology, University of Lübeck and University of Schleswig-Holstein, 23564 Lübeck, Germany; 6Department of Medicine B for Gastroenterology, Hepatology, Endocrinology and Clinical Infectiology, University Hospital Münster, 48149 Münster, Germany; goeranramin.boeckel@ukmuenster.de (G.R.B.); phil-robin.tepasse@ukmuenster.de (P.-R.T.); 7Department of Medicine D for Nephrology and Rheumatology, University Hospital Münster, 48149 Münster, Germany; 8Central Hematology Laboratory, Department of Hematology, Inselspital, Bern University Hospital, University of Bern, 3010 Bern, Switzerland; veraulrike.bacher@insel.ch; 9Department of Infection Control and Infectious Diseases, University Medicine Göttingen (UMG), 37077 Göttingen, Germany; simone.scheithauer@med.uni-goettingen.de

**Keywords:** SARS-CoV-2, COVID-19, COVID-19 vaccination, cancer patients

## Abstract

**Simple Summary:**

This study investigated SARS-CoV-2 infections and their impact on cancer in COVID-19 vaccinated (*n* = 49) and non-vaccinated (*n* = 84) cancer patients. A mild course of COVID-19 was documented more frequently in vaccinated cancer patients (49% vs. 29%), while the incidence of severe and critical courses occurred in approximately one-half of the non-vaccinated patients (22% vs. 42%). In comparison to non-vaccinated patients, admissions to intermediate and intensive care units and the need for non-invasive and invasive respiratory support were reduced by 71% and 50% among vaccinated patients. The median length of hospital stay was 11 days for non-vaccinated and 5 days for vaccinated patients. COVID-19 mortality was reduced by 83% in vaccinated patients. Finally, the median time from SARS-CoV-2 infection to restarting cancer therapy was 12 and 26 days among vaccinated and non-vaccinated groups, respectively. Our results provide evidence for the significant benefits of COVID-19 vaccines for cancer patients.

**Abstract:**

COVID-19 vaccines have become an integral element in the protection of cancer patients against SARS-CoV-2. To date, there are no direct comparisons of the course of COVID-19 infection in cancer patients between the pre- and post-vaccine era. We analyzed SARS-CoV-2 infections and their impact on cancer in COVID-19 vaccinated and non-vaccinated patients from three German cancer centers. Overall, 133 patients with SARS-CoV-2 were enrolled in pre- and post-vaccine eras: 84 non-vaccinated and 49 vaccinated, respectively. A mild course of COVID-19 was documented more frequently in vaccinated patients (49% vs. 29%), while the frequency of severe and critical courses occurred in approximately one-half of the non-vaccinated patients (22% vs. 42%, *p* = 0.023). Particularly, patients with hematologic neoplasms benefited from vaccination in this context (*p* = 0.031). Admissions to intermediate- and intensive-care units and the necessity of non-invasive and invasive respiratory support were reduced by 71% and 50% among vaccinated patients, respectively. The median length of admission was 11 days for non-vaccinated and 5 days for vaccinated patients (*p* = 0.002). COVID-19 mortality was reduced by 83% in vaccinated patients (*p* = 0.046). Finally, the median time from SARS-CoV-2 infection to restarting cancer therapy was 12 and 26 days among vaccinated and non-vaccinated groups, respectively (*p* = 0.002). Although this study does not have enough power to perform multivariate analyses to account for confounders, it provides data on COVID-19 in non-vaccinated and vaccinated cancer patients and illustrates the potential benefits of COVID-19 vaccines for these patients.

## 1. Introduction

The emergence of the novel coronavirus SARS-CoV-2 (severe acute respiratory syndrome coronavirus 2) in 2019 resulted in the global spread of COVID-19, causing a pandemic that has placed an enormous impact on public health. Not surprisingly, cancer patients represent an especially vulnerable cohort of patients due to the immunosuppressive effects of the anticancer therapy and the underlying malignant disease itself [1].

During the first wave of the COVID-19 pandemic. dramatic mortality rates of up to 33% were documented among non-vaccinated cancer patients, which were significantly higher compared to patients without malignancies [2,3]. Subsequently, steady progress has been achieved in pre-exposure prophylaxis and treatment strategies of SARS-CoV-2 aiming at preventing or mitigating severe and critical courses of COVID-19.

The introduction of viral vector- (e.g., AZD1222; Ad26.COV2.S) and mRNA- (e.g., BNT162b2; mRNA-1273) based COVID-19 vaccines [4,5,6,7,8] has changed our perception of the risk of COVID-19 in non-cancer patients. Despite the high protection rate of COVID-19 vaccines in the general population [4], considerable uncertainty still exists regarding their efficacy and the risk of COVID-19 for cancer patients. Fortunately, high seroconversion rates of up to 90% were documented among patients with solid malignancies [9,10]. On the other hand, patients with hematologic malignancies, especially those with B-cell lymphoproliferative neoplasms, exhibited a much lower rate of seropositivity ranging from 40% to 85% following vaccination [11,12,13].

Parallel to vaccination, several treatment strategies emerged aiming at reducing an over-activated host immune response to SARS-CoV-2, as well as inhibiting virus replication and invasion in the host. For instance, remdesivir, an inhibitor of the viral RNA-dependent RNA polymerase, shortened the time to recovery in adults with COVID-19 (10 vs. 15 days for placebo) [14]. Subsequently, several SARS-CoV-2 neutralizing monoclonal antibodies (mABs) such as casirivimab/imdevimab (REGN-CoV-2) and sotrovimab have been approved to prevent or mitigate COVID-19 [15,16]. Notably, sotrovimab has been shown to reduce the risk of hospital admission and death by 79% in high-risk adults with symptomatic COVID-19 infection [15].

Acknowledging all recent successes in the campaign against SARS-CoV-2, it remains to be clarified whether and how the above-mentioned progress has changed the risk of COVID-19 in cancer patients in routine clinical practice.

Before the introduction of vaccination against SARS-CoV-2, we reported diagnostic, clinical, and post-SARS-CoV-2 scenarios in 63 non-vaccinated cancer patients from three German cancer centers [17]. Within that study, 39% of all patients had either severe or critical COVID-19 courses and 13% succumbed to COVID-19. So far, there have been no direct comparisons in cancer patients with COVID-19 between the pre- and post-vaccine eras. To fill this gap, we have extended our previous cohort of 63 patients and analyzed clinical patterns of SARS-CoV-2 infections and their impact on the subsequent clinical course of the malignant disorders in 84 patients before the introduction of the COVID-19 vaccine and compared them to 49 patients from the period after vaccine introduction throughout all variants of the SARS-CoV-2 virus.

## 2. Materials and Methods

### 2.1. Patients

Between March 2020 and February 2022, all patients with active or non-active malignant disease and a SARS-CoV-2 infection confirmed by real-time reverse transcriptase polymerase chain reaction (RT-PCR) presenting in the University Hospital of Münster, the University Hospital of Göttingen, and Departments of Thoracic Oncology and Medical Oncology and Hematology of Franziskus-Hospital Harderberg, Georgsmarienhütte, Germany, were enrolled in the study. From the non-vaccinated cohort, *n*= 63 patients were previously reported from our group [17], and this cohort was now extended with an additional 21 patients. Cancer patients with SARS-CoV-2 were identified either at diagnosis of the SARS-CoV-2 infection or retrospectively in the oncology departments participating in the study. All regular cancer in- and outpatients were screened by COVID-19 questionnaire and a SARS-CoV-2 antigen test performed either in the community or in the participating hospitals before presentation in the oncology departments. Suspected or proven SARS-CoV-2 cases by subsequent RT-PCR were reported to the physicians involved in the study. Infected patients who had cancelled their appointment due to COVID-19 were called at home by physicians and/or investigators to ask about symptoms, medical history, COVID-19, and cancer treatment (if the latter was not available in the medical records). Additionally, the COVID-19 wards were scanned every day for newly admitted cancer patients. The final outcomes of SARS-CoV-2 infection were documented retrospectively at the next follow-up visit following recovery after SARS-CoV-2. For the patients who died post-SARS-CoV-2, data were collected based on the last medical records. All types of COVID-19 vaccines (mRNA and viral vector-based) approved by the Food and Drug Administration (FDA, Silver Spring, USA) and the European Medicines Agency (EMA, 1083 Amsterdam, The Netherlands) were acceptable. Only double-vaccinated patients, with the exception of one patient after a single-dose of Ad26.COV2.S (Johnson and Johnson, Janssen Pharmaceutica N.V., 2340 Beerse, Belgium), were included in the study. The retrospective data analysis was approved by decisions of the local ethics committees (Ethics Committee of the University Medical Center Göttingen № 23/11/21; Ethics Committee of the Lower Saxony Medical Association № 23/11/21; Ethics Committee of the Westphalia-Lippe Medical Association № 2021-817-b-S).

### 2.2. Methods

The presence of SARS-CoV-2 was confirmed by RT-PCR on nasopharyngeal and oropharyngeal swab material in all cases. All diagnostic scenarios were considered: Symptomatic patients or asymptomatic patients within routine and contract tracing screening. Routine SARS-CoV-2 screening referred to in- and outpatients subjected either to PCR- or antigen-based testing without exhibiting any signs or symptoms of COVID-19. Contract tracing was defined as PCR- or antigen-based testing performed for patients with a history of close contact with individuals with laboratory-confirmed or probable COVID-19.

The following commercial SARS-CoV-2 RT-PCR test systems were used: SARS-CoV-2 Cobas^®^ 6800/8800 (Roche, Basel, Switzerland), Alinity m SARS-CoV-2 assay (Abbott, Chicago, IL, USA), Xpert^®^ Xpress SARS-CoV-2 (Cepheid, Sunnyvale, CA, USA), Genesic SARS-CoV-2 (G Healthcare, Chicago, IL, USA), and FTD SARS-CoV-2 assay (Siemens Healthineers Company, 91052 Erlangen, Germany).

### 2.3. Definitions

Patients with active malignant disease were defined as those with either newly diagnosed and yet untreated disease, as well as recurrent, regionally advanced, or metastatic disease for which anticancer treatment had been administered in the preceding six months in any setting (curative, palliative, radical, adjuvant, or neoadjuvant). Accordingly, patients with non-active malignant disease were defined as cancer survivors who met the above-mentioned criteria and were undergoing follow-up surveillance at the time of the SARS-CoV-2 infection. The COVID-19 severity categories were determined according to the WHO guidelines: Asymptomatic, mild (i.e., general symptoms), severe (i.e., need for oxygen supplementation and admission), and critical course (i.e., need for life support therapy) [18]. Lymphocytopenia was documented when a lymphocyte count was below 1.0 × 10^9^/L. The follow-up consisted in monitoring the patients from the first positive SARS-CoV-2 RT-PCR result to the last contact for living patients or death.

### 2.4. Analyzed Data

Both vaccinated and non-vaccinated patients were analyzed considering (A) demographics and cancer data including the most recent anti-cancer therapy and remission status, as well as comorbidities at the time of the first SARS-CoV-2 positive result. In addition, in vaccinated patients, the type of COVID-19 vaccine and the time point of its application, if available, preceding SARS-CoV-2 infection were documented. Furthermore, (B) the course of the SARS-CoV-2 infection including symptoms and their severity and duration, as well as laboratory and radiological findings, and finally, (C) the course of the malignant disorder in the follow-up period, were analyzed. To this end, the impact of SARS-CoV-2 on overall survival (OS) was evaluated and compared between the vaccinated and non-vaccinated groups. Chest imaging at the time point of SARS-CoV-2 infection did not include staging examinations or regular screenings. The patients were assessed for the presence of common comorbidities such as cardiovascular diseases (arterial hypertension, coronary artery disease, chronic heart failure, atrial fibrillation), metabolic diseases (diabetes mellitus, obesity), chronic respiratory and liver diseases, renal insufficiency, cerebrovascular diseases, and autoimmune disorders.

### 2.5. Statistics

Categorical variables were summarized as frequencies and percentages, and continuous variables were summarized as the median and range. Overall survival probability was calculated using the Kaplan–Meier method. Cumulative incidence curves were used for COVID-19 and cancer mortality in a competing risk. The Chi-square test of independence was used to determine the significance of the relationship between vaccination status and the (a) severity of the SARS-CoV-2 course; (b) dyspnea; (c) frequency of chest imaging; (d) admission to intermediate care (IMC)/intensive care unit (ICU); and (e) need for respiratory support. The Mann–Whitney-U test for independent samples was used to determine the significance of the relationship between vaccination status and (a) the length of hospital stay in patients admitted due to COVID-19, and (b) the time from SARS-CoV-2 detection to the next cancer therapy. A *p*-value less than 0.05 was considered statistically significant. Statistical analyses were performed with SPSS, version 26.0 (SPSS, Chicago, IL, USA) and R, version 3.6.2 (https://www.r-project.org/ accessed on 12 December 2019.)

## 3. Results

### 3.1. Characteristics of Non-Vaccinated and Vaccinated Cancer Patients with Positive Test for SARS-CoV-2

A total of 133 patients, 84 (63%) non-vaccinated and 49 (37%) vaccinated, who tested positive for SARS-CoV-2 were included in this study. Patient and disease characteristics at first diagnosis of SARS-CoV-2 are presented in Table 1. The data on vaccination status are presented in Appendix A. In total, 77% (*n* = 37/49) of the vaccinated patients had received mRNA-based COVID-19 vaccines. Another 10% (*n* = 5/10) had either vector- followed by mRNA-based (8%) or vector-based (2%) vaccines only. Of the remaining cases, 11% (*n* = 6/49) were double-vaccinated but lacked data on vaccine type documented in medical records. One patient (2%) received Ad26.COV2.S only one time. Seventeen vaccinated patients (35%) had received a third, mRNA-based, vaccine. The median time from the last vaccination to SARS-CoV-19 detection was 129 days.

The median age at the time of SARS-CoV-2 infection was 58 years in the non-vaccinated cohort compared to 65 years in vaccinated patients. There were more males in both groups with male-to-female ratios of 1.3 and 2.3 for non-vaccinated and vaccinated patients, respectively. Hematologic and solid malignancies were equally distributed in the non-vaccinated group (50% for each), whereas hematologic malignancies were more common in the vaccinated group (65% vs. 35%).

Lymphomas were the most frequent hematologic malignancy in both patient groups (33% non-vaccinated; 47% vaccinated), while lung cancer was the most frequent solid tumor (13% for each). In total, 87% of the non-vaccinated (*n* = 73/84) and 78% of the vaccinated (*n* = 38/49) patients were categorized as active cancer cases, whereas the remaining 12% (*n* = 11/84) and 22% (*n* = 11/49), respectively, were under follow-up surveillance after systemic cancer treatment. At the time of SARS-CoV-2 detection, cytostatic therapy, either alone or in combination with other compounds, was the most common treatment in both groups (57% non-vaccinated; 44% vaccinated), followed by either immunotherapy in the non-vaccinated group or targeted therapy in the vaccinated group (for details see Table 1). Approximately half of all patients had undergone one line of therapy (46% non-vaccinated; 48% vaccinated), while the remaining had undergone two (25% non-vaccinated; 14% vaccinated) or more therapy lines prior to becoming infected with SARS-CoV-2 (see Table 1). At the time of enrollment, the vaccinated patients tended to be in a better state of remission as compared to the non-vaccinated patients: Complete and partial remission (CR + PR) 58% vs. 42%; stable disease (SD) and relapsed/progressive disease: 22% vs. 35%, respectively. Both groups had equal numbers of comorbidities (see Table 1).

### 3.2. Clinical, Laboratory, and Imaging Findings in 133 Cancer Patients with SARS-CoV-2

Laboratory, imaging, and clinical findings in 133 cancer patients with positive SARS-CoV-2 test are presented in Table 2 and Figure 1. The median interval between the most recent cancer treatment and SARS-CoV-2 infection was 13 (non-vaccinated) and 19 (vaccinated) days, respectively. In total, 71% of the patients (*n*= 60/84 non-vaccinated; *n* = 35/49 vaccinated) presented with COVID-19 symptoms while 29% (*n* = 24/24 non-vaccinated; *n* = 14/49 vaccinated) remained asymptomatic during the entire follow-up. In both groups, the most common COVID-19 symptom was cough (68%, *n* = 41/60 non-vaccinated; 89%, *n* = 31/35 vaccinated), followed by fever, dyspnea, gastrointestinal symptoms, and chest pain (Table 2). Of note, dyspnea was somewhat less common in vaccinated (40%; *n* = 14/35) than in non-vaccinated patients (58%; *n* = 35/60), but this difference was not significant (*p* = 0.133). Chest imaging was consequently performed significantly more frequently in non-vaccinated patients (64% vs. 38%) (*p* = 0.005) (Figure 1). The laboratory results including WBCs, CRP, PCT, and LDH did not differ significantly between the two groups (*p* > 0.25) at the time of SARS-CoV-2 detection.

### 3.3. SARS-CoV-2 Infection Course

Data showing the course of the SARS-CoV-2 infection are presented in Table 3 and Figure 1 and Figure 2.

The course of the COVID-19 infection differed significantly between the vaccinated and non-vaccinated groups, with significantly more patients developing mild and significantly fewer having a severe or critical course in the vaccinated group (Table 3 and Figure 2). In detail, 29% (*n* = 24/84) and 49% (*n* = 24/49) had a mild, 22% (*n* = 19/84) and 18% a severe (*n* = 9/49), and 20% (*n* = 17/84) and 4% (*n*= 2/49) a critical course among non-vaccinated and vaccinated patients, accordingly (*p* = 0.023).

Patients with hematologic malignancies had a clear benefit from vaccination: 51% (*n* = 16/31) and 21% (*n* = 9/42) had a mild, 13% (*n* = 4/31) and 21% a severe (*n* = 9/42), and 7% (*n* = 2/31) and 24% (*n* = 10/42) a critical course among vaccinated and non-vaccinated cases (*p* = 0.031), accordingly. Similarly, among patients with solid tumors, there were more mild courses (44% vs. 35%) and no critical courses (0% vs. 17%) in vaccinated vs. non-vaccinated patients, but the differences did not reach significance (Figure 2). In the groups of vaccinated patients, those with hematologic malignancies fared significantly worse than those with solid tumors. Seven percent of them (*n* = 2/31) and none with a solid tumor had a critical COVID-19 course (Figure 2).

The proportions of vaccinated and non-vaccinated patients admitted to the hospital did not differ significantly (71%, *n* = 35/49 vs. 65%, *n* = 55/84). Admission was usually due to COVID-19 symptoms (51%, *n* = 25/49 vaccinated; 39%, n = 33/84 non-vaccinated) followed by anti-SARS-CoV-2 antibody treatment, if available. Asymptomatic SARS-CoV-2 carriers were also admitted for clinical observation if deemed at high-risk, e.g., due to severe lymphocytopenia after chemotherapy (5%, *n* = 4/84 non-vaccinated; 10%, *n* = 5/49 vaccinated). The remaining admissions took place prior to a positive SARS-CoV-2 test on the ward: 21% (*n* = 18/84) non-vaccinated and 10% (*n* = 5/49) vaccinated patients.

Significantly more of the non-vaccinated hospitalized patients had to be transferred to an intermediate-care (IMC) or intensive-care unit (ICU) for further anti-infectious treatment compared to vaccinated patients (31% vs. 11%; *p* = 0.033) (Figure 1). Non-invasive (continuous positive airway pressure or high-flow oxygen therapy) or invasive mechanical respiratory support was necessary for 24% (*n* = 13/55) of the non-vaccinated and 14% (*n* = 435) of the vaccinated patients (*p* = 0.15) (Figure 1). Non-vaccinated patients had a median hospital stay of 11 days compared with 5 days for the vaccinated patients (*p* = 0.002; 95%-CI [2,11]) (see Table 3 for details). Data on lymphocyte counts were available for only 61 of 133 patients with 28 of them presenting with lymphocytopenia. However, this finding did not correlate with the severity of the COVID-19 course (*p* > 0.05). Of all patients, 28 had received anti-CD20 therapy previously, which did not have any impact on the COVID-19 course in comparison to the remaining patients (*p* > 0.05).

Different treatment modalities were used in the non-vaccinated and vaccinated groups ranging from monitoring (30% non-vaccinated; 19% vaccinated) or only symptomatic treatment (23% non-vaccinated; 26% vaccinated) to specific antibody therapy with more than one employed modality possible (47% non-vaccinated; 55% vaccinated). Antibiotics were used more frequently in the non-vaccinated group (61% vs. 12%), while specific COVID-19 treatments were applied with a similar frequency in both groups (47% vs. 55%). Specific therapeutics were antibody-based agents (52% non-vaccinated; 77% vaccinated), virostatics (26% non-vaccinated; 15% vaccinated), dexamethasone (11% non-vaccinated; 19% vaccinated), and serine protease/JAK-inhibitors (3% non-vaccinated; 4 % vaccinated). For more details on specific therapies, see Table 3.

### 3.4. Outcomes of SARS-CoV-2 Infection with Impact on Cancer Course

The outcomes and impact of the SARS-CoV-2 infection on the subsequent course of the malignant disorder are presented in Table 3 and Figure 3A–C. The median interval from SARS-CoV-2 positivity to the last follow-up was five weeks (range, 0–62). At the last follow-up, 23% (*n* = 19/84) of the non-vaccinated and 6% (*n* = 3/49) of the vaccinated patients had died resulting in estimated three-month overall survival rates after the SARS-CoV-2 infection of 75% and 89%, respectively (Figure 3). In particular, COVID-19 was the cause of death in 53% (*n* = 10/19) of the non-vaccinated and 33% (*n* = 1/3) of the vaccinated patients, while a relapsed/progressive malignancy was the cause in 47% (*n* = 9/19) and 67% (n = 2/3), respectively (Figure 3). Of the non-vaccinated patients who succumbed to COVID-19, 50% (*n* = 5/10) had a hematologic (B-cell lymphoma, *n* = 4/10; AML, *n* = 1/10) and 50% a solid malignancy (*n* = 5/10). Lung cancer was the most common solid tumor (40%; *n* = 4/10). Both vaccinated patients who succumbed to COVID-19 had a history of B-cell lymphoma. COVID-19 and cancer-associated mortality occurred with a median time of 2.5 and 4.5 weeks from the first diagnosis of SARS-CoV-2 infection, accordingly.

In total, 103 patients (69 non-vaccinated; 34 vaccinated) had ongoing or scheduled cancer therapy before they contracted SARS-CoV-2. Treatment was delayed in 84% (*n* = 58/69) and 65% (*n* = 22/34) of them, respectively (Figure 3). The median time from virus detection to the next therapy was 26 and 12 days, respectively, within non-vaccinated and vaccinated groups (*p* = 0.002; 95%-CI [8,19]). Of the non-vaccinated patients, 31% (*n* = 18/58) developed a progressive disease (PD) during the SARS-CoV-2 infection. In the group of vaccinated patients, 15% (3/20) had PD under the described circumstances. The information on two patients was missing.

## 4. Discussion

Here, we comprehensively analyzed clinical scenarios of SARS-CoV-2 infection in cancer patients in the periods just before and after the introduction of COVID-19 vaccines. The aim was to provide real-life data on the progress in SARS-CoV-2 management and the prognosis of these vulnerable patients in the COVID-19 pandemic. To our knowledge, this is the first multicenter study evaluating the outcomes of vaccinated and non-vaccinated COVID-19 patients.

Our patient cohort encompassed the entire spectrum of solid and hematologic neoplasms, as well as the cancer treatment modalities in outpatient and inpatient care settings. The patients’ characteristics reflected everyday clinical practice in so far as more than half of the patients were 60 years or older, had received one or more therapy lines, and had one or more comorbidities.

Overall, nearly one-third of the patients in both groups were asymptomatic. With regard to the COVID-19 symptoms, fever, cough, and dyspnea were the COVID-19 symptoms recorded most frequently in our patients among both groups, although dyspnea was documented 31% less frequently in vaccinated patients than in non-vaccinated patients. Accordingly, a mild COVID-19 course was almost 69% more frequent in vaccinated patients. In non-vaccinated patients, the frequency of severe and critical courses was 42%, which is similar to that in other large multicenter studies from the pre-vaccine era [2,20,21]. On the other hand, the incidence of severe and critical courses in vaccinated patients was only approximately one-quarter (22%) in our study. In particular, a critical COVID-19 course in vaccinated patients was documented 80% less than in non-vaccinated patients.

Fewer cases of severe COVID-19 in vaccinated patients consequently meant less pressure on the healthcare system. Vaccinated patients required significantly less frequent chest imaging, generally required less extensive and invasive therapy, and had a significantly shorter in-hospital stay. However, Schmidt et al. found no differences in the severity of COVID-19, frequency of admission to the intensive-care unit, and the need for mechanical ventilation between vaccinated and non-vaccinated patients on a time scale preceding the emergence of the Omicron (B1.1.529) variant [22]. Thus, further large-scale studies addressing these aspects are necessary in the near future.

In parallel to the development and introduction of vaccines, significant progress has been achieved in the field of specific COVID-19 therapy [14,15,16]. Recently, we have demonstrated SARS-CoV-2 clearance and reduction for oxygen supplementation in high-risk patients with hematologic malignancies following the application of mABs neutralizing SARS-CoV-2 [23]. Within our study, the proportion of patients that received neutralizing mABs binding to the SARS-CoV-2 spike protein was three-fold lower among non-vaccinated cases in comparison to vaccinated ones, which could also have had an impact on COVID-19 course within both groups. Thus, the combination of both COVID-19 vaccines preceding, and specific COVID-19 therapy following, SARS-CoV-2 infection can reduce the severity of COVID-19 in cancer patients despite a significantly weakened immune system per se. 

Overall mortality averaged 23% and 6% in our analysis for non-vaccinated and vaccinated patients, resulting in an estimated three-month overall survival rate after a SARS-CoV-2 infection of 75% and 89%, respectively. Yet, COVID-19 mortality was 83% less in the group of vaccinated patients. In comparison to that, two previous studies reported overall COVID-19 mortality rates of approximately 13% in 56 and 113 mostly double-vaccinated patients, respectively [22,24]. However, the two studies did not comment on the use of mABs.

Notably, one-third of our patients had received booster vaccinations prior to acquiring the SARS-CoV-2 infection, and this could have also influenced the outcomes [19]. Since November 2021, the B.1.1.529 (omicron) subgroup of COVID-19 has rapidly become the dominant SARS-CoV-2 globally. Recently, Fendler et al. found that a third vaccine dose increased the neutralizing antibody titer (nAbT) against the omicron variant in patients with cancer. Indeed, omicron nAbT was detectable in 37% and 90%, as well as in 15% and 56%, of patients with solid and hematologic malignancies before and after a third vaccination, respectively [25]. Hence, booster injections are of particular importance for cancer patients and should be repeated in patients failing to develop a serological response. In addition, the omicron variant itself is known to cause fewer critical courses, which may have contributed to the drastic differences within our patient cohort, since approximately 80% of the vaccinated patients had COVID-19 beginning in autumn 2021.

Considering the impact of SARS-CoV-2 infection on subsequent cancer management, vaccinated patients benefited from having a shorter treatment delay compared to non-vaccinated patients. Indeed, the median time from diagnosis of a SARS-CoV-2 infection to restarting therapy was only half that in the vaccinated group.

Despite vaccination, approximately one-fifth of the patients in our study still had a severe or critical course, highlighting the fact that cancer patients remain extremely vulnerable in the COVID-19 pandemic. Particularly, hematologic patients are at risk of COVID-19. Along this line, Mittelman et al. demonstrated an increased relative risk for hospitalization (3.13), severe COVID-19 (2.27), and death due to COVID-19 (1.67) among 32,516 double-vaccinated hematologic patients in comparison to a matched control accounting for 32,516 individuals without hematologic neoplasms [26]. Many questions have thus arisen about the impact of COVID-19 vaccines in cancer patients under ongoing, particularly B-cell-depleting, therapy. Recently, Liebers et al. investigated the neutralizing antibody capacity and T-cell responses following double COVID-19 vaccination in 80 lymphoma patients who had received anti-CD20 treatment. In accordance with other reported data, 41% of those patients achieved seroconversion [27,28,29]. Interestingly, 70% of the seroconverted patients exhibited a T-cell response, while 50% of patients without seroconversion still demonstrated a T-cell response. As opposed to the serological response, vaccine-induced T-cell responses were not dependent on the interval between the last anti-CD20 treatment and the vaccination [30]. Yet, T-cell responses are known to be important for viral clearance [30], and it is still possible that they are sufficient to enable a mild course of COVID-19. Consequently, COVID-19 vaccination might be effective even in B-cells lymphodepleted cancer patients due to T-cell immunity.

Addressing the possible bias of this retrospective analysis it should be highlighted that our study does not have enough power to perform multivariate analyses to account for confounders. The latter should certainly be considered, as our study has several limitations. First, our pilot study only included a relatively small number of vaccinated patients. Second, the proportion of vaccinated patients with no therapy due to first diagnosis and/or a “watch and wait” strategy, and aftercare following systemic cancer treatment was two-fold higher in comparison to non-vaccinated patients. This may have had an impact on the differences in COVID-19 courses documented among vaccinated and non-vaccinated patients in our study. Thus, the received data cannot be generalized to all cancer patients and should be interpreted in daily practice depending on clinical settings. In addition, neither post-vaccination titers nor T-cell-mediated immunity were determined in our study. Finally, one can assume that vaccinated patients might have been infected more often with the omicron variant, which is not as aggressive as earlier SARS-CoV-2 variants.

## 5. Conclusions

Based on our data and the current literature, we summarize the following key-points:(1)The likelihood of a milder COVID-19 course is much greater in vaccinated than in non-vaccinated cancer patients.(2)COVID-19 vaccines reduce the number of admissions to intermediate/intensive care units, reduce the need for non- and invasive respiratory support, and shorten the length of in-hospital stay following COVID-19 in cancer patients.(3)COVID-19 vaccines enable significantly earlier resumption of cancer therapies after SARS-CoV-2 infection(4)Despite vaccination, up to 20% of all cancer patients remain at high risk for a severe or critical COVID-19 course, which mandates a further urgent improvement in pre- and post-SARS-CoV-2 exposure strategies.

## Figures and Tables

**Figure 1 cancers-14-03746-f001:**
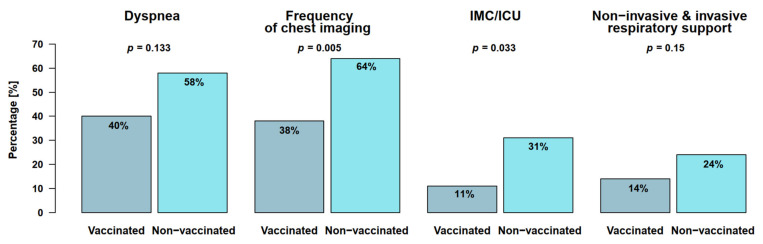
Clinical, diagnostic, and treatment aspects of COVID-19 course among non- and vaccinated cancer patients of the study.

**Figure 2 cancers-14-03746-f002:**
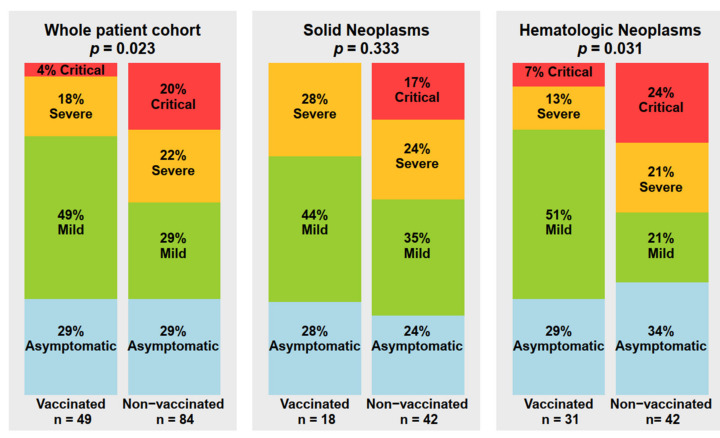
Severity of SARS-CoV-2 infection in non- and vaccinated cancer patients of the study.

**Figure 3 cancers-14-03746-f003:**
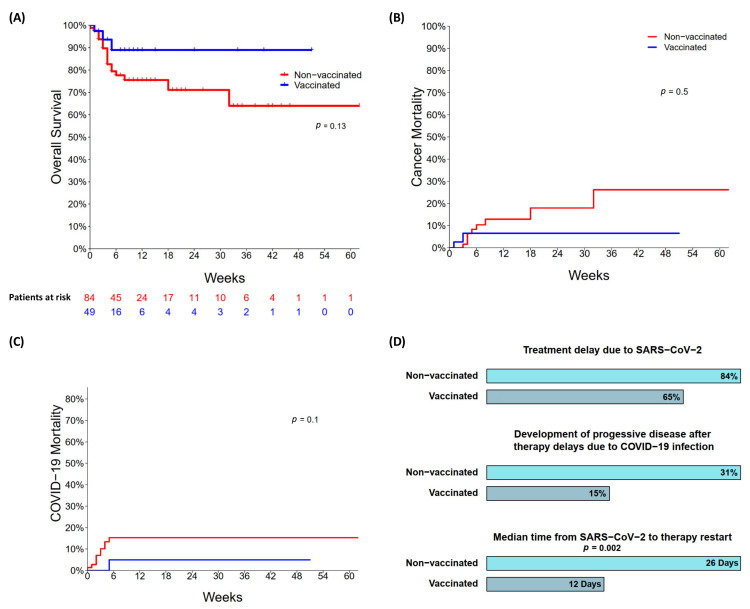
(**A**) Overall survival among non- and vaccinated patients; (**B**) cancer mortality among non- and vaccinated patients; (**C**) COVID-19 mortality among non- and vaccinated patients; (**D**) treatment delay, development of progressive disease, and median time from SARS-CoV-2 infection to therapy restart among non- and vaccinated patients of this study.

**Table 1 cancers-14-03746-t001:** Clinical characteristics of 133 cancer patients at the time of SARS-CoV-2 infection. M, male; F, female; n, number; ALL, acute lymphoblastic leukemia; AML, acute myeloid leukemia; PNH, paroxysmal nocturnal hemoglobinuria; ITP, immune thrombocytopenia; MDS, myelodysplastic syndrome.

Parameter	Non-Vaccinated, *n* = 84		Vaccinated, *n* = 49		*p*-Value *
Vaccination status	84/133	63%	49/133	37%	
Gender (M/F), *n* (ratio)	47/37	56%/44%	34/15	69%/31%	0.127
Median age, years (range)	58	19–85	65	24–85	0.060
Cancer entities, *n*	*n* = 84		*n* = 49		
Solid tumors, *n* (%)	42/84	50%	17/49	35%	0.088
Lung cancer Breast cancer Gastrointestinal cancer Sarcoma Gynecologic cancer Endocrine cancer Brain cancer Melanoma Urogenital cancer Ear, nose, and throat carcinoma	11 8 8 4 2 3 1 1 3 1	13% 10% 10% 4% 2% 4% 1% 1% 4% 1%	6 1 3 2 - 1 - - 3 1	13% 2% 6% 4% - 2% - - 6% 2%	
Hematologic malignancies, ***n*** (%)	42/84	50%	32/49	65%	0.088
**Lymphoma** ***-B-cell lymphoma*** ***-T-cell lymphoma*** **Multiple myeloma** Acute leukemia ***-ALL*** ***-AML*** Others	28 27 1 2 11 6 5 1	33% 32% 1% 2% 14% 8% 6% 1%	23 21 - - 7 2 5 2	47% 43% - - 14% 4% 10% 4%	
Cancer treatment preceding SARS-CoV-2 positivity, *n* (%)					0.280
Cytostatic therapy ***-cytostatic agents only*** ***-combined with immunotherapy*** ***-combined with targeted therapy*** ***-combined with radiotherapy*** Immunotherapy ***-immunotherapy mono-regimen*** ***-combined with radiotherapy*** Targeted therapy Surgery Hormonal therapy Radiotherapy No therapy yet due to first diagnosis or “watch and wait” strategy Aftercare following systemic cancer treatment	47 27 13 4 3 10 7 3 4 4 1 2 5 11	57% 32% 16% 5% 4% 12% 8% 4% 5% 5% 1% 2% 6% 12%	21 6 11 2 2 1 1 - 6 2 - - 8 11	44% 13% 23% 4% 4% 2% 2% - 12% 4% - - 16% 22%	
Number of therapy lines at SARS-CoV-2 positivity, *n* (%)					0.184
No therapy yet	5	6%	7	14%	
1 therapy line	38	46%	23	48%	
2 therapy lines	21	25%	7	14%	
3 therapy lines	4	5%	4	8%	
≥4 therapy lines	16	18%	8	16%	
Remission status at SARS-CoV-2 positivity, *n* (%)					0.169
Complete remission	19	23%	14	29%	
Partial remission	16	19%	14	29%	
Stable disease	13	15%	5	10%	
Relapsed/progressive disease	25	30%	6	12%	
Not yet assessed	11	13%	10	20%	
Comorbidities, ***n***(%)	61/84	73%	40/49	82%	0.245
1 comorbidity	22	36%	14	35%	
2 comorbidities	20	33%	12	30%	
3 comorbidities	13	21%	8	20%	
≥4 comorbidities	6	10%	6	15%	

* Mann-Whitney U Test.

**Table 2 cancers-14-03746-t002:** Laboratory, imaging, and clinical findings in 133 cancer patients with SARS-CoV-2. RT-PCR, real-time reverse transcriptase polymerase chain reaction; *n*, number; FU, follow-up; CT, computer tomography; WBCs, white blood cells; CRP, c-reactive protein; PCT, procalcitonin; LDH, lactate dehydrogenase; pts, patients.

Parameter	Non-Vaccinated, *n* = 84		Vaccinated, *n* = 49	
**Median time from last cancer treatment to first positive SARS-CoV-2 RT-PCR test ^§^, days**	13		19	
Patients with COVID-19 symptoms during the whole FU	60/84	71%	35/49	71%
**COVID-19 symptoms** (multiple answers possible), ***n*** **(%)**	60/84	71%	35/49	71%
Cough	41/60	68%	31/35	89%
Fever	40/60	67%	19/35	54%
Dyspnea	35/60	58%	14/35	40%
Gastrointestinal symptoms	9/60	15%	7/35	20%
Chest pain	6/60	10%	3/35	9%
**Laboratory results at SARS-CoV-2 detection**				
WBCs × 10^9^/L, median (range) *	5.7	0.3–42.9	5,4	0.4–47.6
CRP, mg/dl, median (range) **	21.	0.7–308	16	0.3–273
PCT, ng/mL, median (range) ***	0.17	0.0–84,3	0,1	0–3.1
LDH, U/L, median (range) ****	277	86–2830	292	125–2117
Lymphocytopenia (<1.0 × 10^9^/L) (data available for 46 unvaccinated and 15 vaccinated patients)	20/46	43%	8/15	53%
**Chest imaging for diagnosing COVID-19, *n* (%)** (data available for 77 unvaccinated and 42 vaccinated patients) multiple answers possible				
X-ray	27/49	55%	7/16	44%
CT	31/49	63%	9/16	56%
Ultrasound	3/49	6%	-	-
**Total number of patients underwent imaging for diagnosing COVID-19,** ** *n* ** **(%)**	49/77	64%	16/42	38%
**Imaging results,** ** *n* ** **(%)**				
Signs of pneumonia	32/49	65%	10/16	63%
No indication of pneumonia	17/49	35%	6/16	37%
** ^§^ ** **Confirmed secondary bacterial infections in COVID-19 cases,** ** *n* ** **(%)**	12/84	14%	7/49	14%

* available for 54/84 unvaccinated and 35/49 vaccinated patients; ** available for 47/84 unvaccinated and 31/49 vaccinated patients *** available for 24/84 unvaccinated and 14/49 vaccinated patients **** available for 46/84 unvaccinated and 30/49 vaccinated patients. ^§^ PCR test systems used: University Hospital Münster: SARS-CoV-2 Cobas^®^ 6800/8800 (Roche, Basel, Switzerland), Alinity m SARS-CoV-2 assay (Abbott, Chicago, IL, USA), Xpert^®^ Xpress SARS-CoV-2 (Cepheid, Sunnyvale, CA, USA). University Hospital Göttingen: SARS-CoV-2 Cobas^®^ 6800/8800 (Roche, Basel, Switzerland), Alinity m SARS-CoV-2 assay (Abbott, Chicago, IL, USA), Xpert^®^ Xpress SARS-CoV-2 (Cepheid, Sunnyvale, CA, USA), Genesic SARS-CoV-2 (G Healthcare, Chicago, IL, USA).Franziskus-Hospital Harderberg, Niels-Stensen-Kliniken, Georgsmarienhütte: FTD SARS-CoV-2 assay (Siemens Healthineers Company, Germany).^§^ defined as proven either in trachealbronchial fluid, blood or urine culture bacterial infection emerged within the course of COVID-19.

**Table 3 cancers-14-03746-t003:** Outcomes of SARS-CoV-2 infection in 133 cancer patients. *n*, number; IMC, intermediate care; ICU, intensive care unit; PD, progressive disease; FU, follow-up.

Parameter	Non-Vaccinated (*n* = 84)		Vaccinated (*n* = 49)	
**Severity of SARS-CoV-2 infection, *n* (%)**				
Asymptomatic SARS-CoV-2 course	24/84	29%	14/49	29%
COVID-19 ***-mild course*** ***-severe course*** ***-critical course***	60 24/84 19/84 17/84	71% 29% 22% 20%	35 24/49 9/49 2/49	71% 49% 18% 4%
**Patient care, *n* (%)**				
Outpatients	29	35%	14	29%
Inpatients ***-admitted because of COVID-19 symptoms*** ***-routine admission prior to positive SARS-CoV-2 test, of them:*** ***-admission of asymptomatic SARS-CoV-2 cases for observation/ therapy***	55/84 33/84 18/84 4/84	65% 39% 21% 5%	35/49 25/49 5/49 5/49	71% 51% 10% 10%
**Requiring wards during hospitalization, *n* (%)**	**55/84**	**65%**	**35/49**	**71%**
General ward	38/55	69%	31/35	89%
IMC/ICU	17/55	31%	4/35	11%
**Non- and invasive respiratory support, *n* (%)**	13/55	24%	4/35	14%
Non-invasive respiratory support ***-high-flow oxygen*** ***-non-invasive mechanical ventilation*** **Invasive mechanical ventilation**	8/55 6/55 2/55 5/55	15% 11% 4% 9%	2/35 2/35 - 2/35	7% 7% - 7%
**Median length of hospital stay of patients admitted because of COVID-19, days, range**	11	1–100	5	1–33
**Treatment modalities related to SARS-CoV-2 infection, *n* (%) among data available for 81/84 unvaccinated and 47/49 vaccinated patient, (multiple answers possible)**				
No therapy at all	24/81	30%	9/47	19%
Symptomatic therapy only	19/81	23%	12/47	26%
Specific COVID-19 therapy	38/81	47%	26/47	55%
** *-virostatics: remdesevir, molnupiravir* ** ** *-dexamethasone* ** ** *Antibody-based therapy* ** ** *-anti-IL-6 receptor/anti-IL-6 antibody* ** ** *-convalescent plasma* ** ** *-antibody against spike protein* ** ** *JAK-Inhibitor/ serine protease inhibitors* **	10/38 4/38 20/38 2/38 8/38 10/38 1/38	26% 11% 52% 5% 21% 26% 3%	4/26 5/26 20/26 - - 20/26 1/26	15% 19% 77% - - 77% 4%
Antibiotic therapy	23/38	61%	3/26	12%
**Treatment delay due to SARS-CoV-2 among 69/84 unvaccinated and 34/49 vaccinated patients with ongoing or planned cancer therapy before, *n* (%)**	58/69	84%	22/34	65%
**Median time from SARS-CoV-2 diagnosis to restart of therapy**	26	0–70	12	0–56
**Median time from SARS-CoV-2 detection to last follow-up, weeks, range within both groups**	5 (0–62)			
**Remission status at last follow-up, *n* (%) ****				
Complete remission	22	26%	11	25%
Partial remission	16	19%	15	34%
Stable disease	24	29%	10	23%
Relapse/progressive disease	22	26%	8	18%
**Survival status at last follow-up, *n* (%)**				
Alive	65	77%	46	94%
Dead	19	23%	3	6%
**Causes of death**				
COVID-19	10/84	12%	1/49	2%
Relapsed/refractory malignancy	9/84	11%	2/49	4%

** only available for 47/49 vaccinated patients, 3 with first diagnosis.

## Data Availability

The data presented in this study are available on request from the corresponding author.

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
