# Peer review of "Clinical Post-SARS-CoV-2 Infection Scenarios in Vaccinated and Non-Vaccinated Cancer Patients in Three German Cancer Centers: A Retrospective Analysis"

_cancers, 2022, doi:10.3390/cancers14153746_

Round 1

Reviewer 1 Report

Original article about the effect of the vaccination to reduce the severity of SARS-CoV-2 among cancer patients and the influence of the infection in the delay about their cancer treatments whether systemic therapy like chemotherapy,  immuntherapy or radiotherapy.  very good written article,  well good job

Author Response

Reviewer 1:

Original article about the effect of the vaccination to reduce the severity of SARS-CoV-2 among cancer patients and the influence of the infection in the delay about their cancer treatments whether systemic therapy like chemotherapy, immunotherapy or radiotherapy.  Very good written article, well good job.

Authors: We appreciate the positive feedback from the reviewer and would like to thank him/her for diligent work on our manuscript.

Reviewer 2 Report

Dear authors,

Along the manuscript, potential confounders and moderators of the association between vaccination and clinical outcomes of SARS-CoV-2 infection were presented, namely, seroconversion after vaccination (much lower in haematologic cancers) and active malignant disease. In table 1, differences in the characteristics of the vaccinated and non-vaccinated patients are notable (no statistical test for the comparison is presented) but results are presented as crude values.  

Without any technique for controlling for confounders, it is not possible to appreciate the results.

The manuscript does not provide information on how patients were identified retrospectively (cancer register of the hospitals? data from the oncology department? list of patients followed by doctors envolved in study?). The diagnosis of SARS-CoV-2 infection is described but not how this information was retrieved: the diagnostic test is always performed at the hospitals envolved in the study? What about tests performed in the community? 

Patients with non-active malignant disease represent a very heterogenous population (early and long survivors) and those who presented at the hospitals during the study period may have different characteristics compared to those who did not, considering the possibilty of clinical follow up at primary care and remote consultation. The results may be biased and certainly can not be generalized to all patients with non-active malignant disease.

Author Response

Reviewer 2:

Along the manuscript, potential confounders and moderators of the association between vaccination and clinical outcomes of SARS-CoV-2 infection were presented, namely, seroconversion after vaccination (much lower in haematologic cancers) and active malignant disease. In table 1, differences in the characteristics of the vaccinated and non-vaccinated patients are notable (no statistical test for the comparison is presented) but results are presented as crude values. 

  1. Without any technique for controlling for confounders, it is not possible to appreciate the results.

Authors: We totally agree with the reviewer and would like to thank him/her for this important point. We calculated the p-values for the parameters listed in the table 1. We found no significant differences between both groups considering age, gender, cancer entities, applied therapies and remission status. We added the values in the table 1.

  1. The manuscript does not provide information on how patients were identified retrospectively (cancer register of the hospitals? data from the oncology department? list of patients followed by doctors envolved in study?). The diagnosis of SARS-CoV-2 infection is described but not how this information was retrieved: the diagnostic test is always performed at the hospitals envolved in the study? What about tests performed in the community? 

Authors: Indeed, this is a crucial methodological aspect, which should be mentioned in the article.

Cancer patients with SARS-CoV-2 were identified either at diagnosis of SARS-CoV-2 infection or retrospectively in the oncology departments participating in the study. All regular cancer in- and outpatients were screened by COVID-19 questionnaire and SARS-CoV-2 antigen test performed either in the community or in the participating hospitals before presentation in the oncology departments. Suspected or proven SARS-CoV-2 cases by subsequent RT-PCR were reported to the physicians involved in the study. Infected patients who had cancelled their appointment due to COVID-19 were called at home by physicians and/or investigators to ask about symptoms, medical history, COVID-19 and cancer treatment (if the latter was not available in the medical records). Additionally, the COVID-19 wards were scanned every day for newly admitted cancer patients. The final outcomes of SARS-CoV-2 infection were documented retrospectively at the next follow-up visit following recovery after SARS-CoV-2. For the patients who died post-SARS-CoV-2 the data were collected based on the last medical records. We added this information in lines 117-129.

  1. Patients with non-active malignant disease represent a very heterogenous population (early and long survivors) and those who presented at the hospitals during the study period may have different characteristics compared to those who did not, considering the possibilty of clinical follow up at primary care and remote consultation. The results may be biased and certainly can not be generalized to all patients with non-active malignant disease.

Authors: We agree with the reviewer that our patient cohort was very heterogenous and this should be considered when analyzing the results among non- and vaccinated patients. At the same time, the reported patient collective represents the real-world situation that oncologists face daily during the COVID-19 pandemic. These aspects have now been stronger emphasized in the manuscript. Considering the extent of primary care and remote consultation, all outpatient cases with non-active malignancies were consulted by responsible oncologists either in the hospital during COVID-19 treatment (e.g. application of neutralizing antibodies), remotely by phone and/or additionally during the next follow-up visit following recovery after COVID-19. We commented on these points as follows (lines 496-502):

“Second, the proportion of vaccinated patients with no therapy due to first diagnosis and/or a “watch and wait” strategy, and aftercare following systemic cancer treatment was two-fold higher in comparison to non-vaccinated patients. This may have had an impact on the difference of COVID-19 course documented among vaccinated and non-vaccinated patients in our study. Thus, the received data cannot be generalized to all cancer patients and should be interpreted in the daily practice depending on clinical settings.” 

Reviewer 3 Report

Overview and general recommendation:

Here, the author studied the SARS-CoV-2 infections and their impact on cancer in COVID-19 vaccinated and non-vaccinated patients, and suggesting there is a significant benefits of COVID-19 vaccines for cancer patients. And it is interesting work. However, the retrospective cohort study should more accurate, the authors listed that both all patients with active or non-active malignant disease and all types of COVID-19 vaccines are not precious for unified analysis. Moreover, how the authors reduce or avoid possible bias in the results, please figure out them. In addition, the benefits of vaccine whether on solid neoplasia or hematologic neoplasia is not clear in Abstracts.

 2.1. Major comments:

1.     Although the Chest imaging was consequently performed significantly more frequently in non-vaccinated patients (64% vs. 38%) (p=0.005) line 230 to line231 of page 6, however, the author did not indicate whether all 133 patients included patients with lung cancer, therefore, the results should not be accurate enough.

2.     The authors concluded that the benefits of COVID-19 vaccines for hematological neoplasms, however, the vaccinated patients with hematological neoplasms, compared with vaccinated matched impaired immunity controls, had an increased risk of documented infections. And how the authors explain them. (See from Blood. 2022 Mar 10;139(10):1439-1451. doi: 10.1182/blood.2021013768.)

2.2. Minor comments:

1.     There are no relative risk and 95% CI analysis results in all the results part. Suggest the authors add them respectively.

Author Response

Reviewer 3:

Overview and general recommendation:

Here, the author studied the SARS-CoV-2 infections and their impact on cancer in COVID-19 vaccinated and non-vaccinated patients, and suggesting there is a significant benefits of COVID-19 vaccines for cancer patients. And it is interesting work. However, the retrospective cohort study should more accurate, the authors listed that both all patients with active or non-active malignant disease and all types of COVID-19 vaccines are not precious for unified analysis. Moreover, how the authors reduce or avoid possible bias in the results, please figure out them. In addition, the benefits of vaccine whether on solid neoplasia or hematologic neoplasia is not clear in Abstracts.

Authors: We would like to thank the reviewer for her/his important improvement suggestions. As far as our article aims to reflect COVID-19 courses among cancer patients from daily clinical practice, we deliberately included all types of cancer patients, e.g. with and without active malignant disease, and all types of vaccines oncologists deal with during the COVID-19 pandemic. We totally agree with the reviewer that the heterogenous patient collective biases the results of our study. To overcome this bias, we added p-values in the Table 1 (baseline characteristics of non- and vaccinated patients). Moreover, we highlight the fact that by analyzing results of the study possible bias should be considered (lines 496-502): “Second, the proportion of vaccinated patients with no therapy due to first diagnosis and/or a “watch and wait” strategy, and aftercare following systemic cancer treatment was two-fold higher in comparison to non-vaccinated patients. This may have had an impact on the difference of COVID-19 course documented among vaccinated and non-vaccinated patients in our study. Thus, the received data cannot be generalized to all cancer patients and should be interpreted in the daily practice depending on clinical settings.”

We also added the information considering the benefits of vaccines (whether on solid neoplasia or hematologic neoplasia) in the abstract: “A mild course of COVID-19 was documented more frequently in vaccinated patients (49% vs. 29%), while the frequency of severe and critical courses occurred in approximately one half of the non-vaccinated patients (22% vs. 42%, p=0.023). Particularly, patients with hematologic neoplasms benefited from vaccination in this context (p=0.031).”  

Round 2

Reviewer 2 Report

Dear authors,

In the discussion, the limitations of the study are well presented, namely, the differences in the administration of specific treatment for COVID-19 and the prevalence of infections with the omicron variant between the pre- and post-vaccine eras; the limited generazibility of the results to the overall population of patients with non-active cancer; the small sample size and differences in cancer type between vaccinated and non-vaccinated cancer patients. 

Therefore, conclusions in the abstract and main text should reflect that the study has not enough power to perform multivariate analyses to account for confounders. The results do not provide robust evidence to of the statements in the conclusions.

I suggest to add "(pre- and post-vaccine eras, respectively)" at the end of the sentence "Overall, 133 patients with SARS-CoV-2 were enrolled: 49 vaccinated and 84 non-vaccinated" of the abstract, to highlight that it is not only the vaccination status that differs between the groups but also knowledge on treatment of COVID-19 and prevalence of omicron infections. 

Please, revise the p value in Table 1, regarding Solid tumors: I calculated a p value of 0.088 instead of p=0.740, and also for Hematologic malignancies, I obtained a p value of 0.088 instead of p=0.670.

Also, please correct these numbers:

- line 346 " 88% of the non-vaccinated (n=73/84)": 73/84=86.9%

- table 3: "Specific COVID-19 therapy 38/81 45%": 38/81=46.9%

Author Response

Reviewer 2:

Dear authors,

In the discussion, the limitations of the study are well presented, namely, the differences in the administration of specific treatment for COVID-19 and the prevalence of infections with the omicron variant between the pre- and post-vaccine eras; the limited generazibility of the results to the overall population of patients with non-active cancer; the small sample size and differences in cancer type between vaccinated and non-vaccinated cancer patients. 

  1. Therefore, conclusions in the abstract and main text should reflect that the study has not enough power to perform multivariate analyses to account for confounders. The results do not provide robust evidence to the statements in the conclusions.

Authors: We would like to thank the reviewer for her/his very thorough review of our manuscript. Indeed, due to the proposals made by the reviewer the manuscript has improved significantly. We agree with the reviewer that it is important to highlight the fact that our results still do not provide robust evidence for all vaccinated patients and should be not generalized. We added the points addressed by the reviewer as follows:

Abstract: “Although this study has not enough power to perform multivariate analyses to account for confounders, it provides data on COVID-19 in non-vaccinated and vaccinated cancer patients, and illustrates the potential benefits of COVID-19 vaccines for these patients.”

Main text, lines 497-499: “Addressing possible bias of this retrospective analysis, it should be highlighted that our study has not enough power to perform multivariate analyses to account for confounders. The latter should be certainly considered as far as our study has several limitations.”

  1. I suggest to add "(pre- and post-vaccine eras, respectively)" at the end of the sentence "Overall, 133 patients with SARS-CoV-2 were enrolled: 49 vaccinated and 84 non-vaccinated" of the abstract, to highlight that it is not only the vaccination status that differs between the groups but also knowledge on treatment of COVID-19 and prevalence of omicron infections. 

Authors: Indeed, this is a very good point. We added this fact in the abstract as follows:  Overall, 133 patients with SARS-CoV-2 were enrolled in pre- and post-vaccine eras: 84 non-vaccinated and 49 vaccinated, respectively.

  1. Please, revise the p value in Table 1, regarding Solid tumors: I calculated a p value of 0.088 instead of p=0.740, and also for Hematologic malignancies, I obtained a p value of 0.088 instead of p=0.670.

Authors: We would like to apologize for these mistakes and thank again the reviewer for her/his very thorough review. We corrected p values as proposed.

  1. Also, please correct these numbers:

- line 346 " 88% of the non-vaccinated (n=73/84)": 73/84=86.9%

- table 3: "Specific COVID-19 therapy 38/81 45%": 38/81=46.9%

Authors: We corrected the mentioned numbers as proposed.  

Once more, we would like to express our thanks to the reviewer. We are hopeful that our revised manuscript is now suitable for publication in Cancers.

Yours sincerely,

Evgenii Shumilov, MD

Annalen Bleckmann, MD

(on behalf of all authors)